# Involuntary Treatment for Child and Adolescent Anorexia Nervosa—A Narrative Review and Possible Advances to Move Away from Coercion

**DOI:** 10.3390/healthcare11243149

**Published:** 2023-12-12

**Authors:** Ingar M. Zielinski-Gussen, Beate Herpertz-Dahlmann, Brigitte Dahmen

**Affiliations:** Department of Child and Adolescent Psychiatry, Psychotherapy and Psychosomatics, University Hospital RWTH Aachen, 52074 Aachen, Germany

**Keywords:** anorexia nervosa, child and adolescent, involuntary treatment, home treatment

## Abstract

Background: Children and adolescents with psychiatric disorders frequently experience hospital treatment as coercive. In particular, for patients with severe anorexia nervosa (AN), clinical and ethical challenges often arise if they do not voluntarily agree to hospital admission, often due to the ego-syntonic nature of the disorder. In these cases, involuntary treatment (IVT) might be life-saving. However, coercion can cause patients to experience excruciating feelings of pressure and guilt and might have long-term consequences. Methods: This narrative review aimed to summarize the current empirical findings regarding IVT for child and adolescent AN. Furthermore, it aimed to present alternative treatment programs to find a collaborative method of treatment for young AN patients and their families. Results: Empirical data on IVT show that even though no inferiority of IVT has been reported regarding treatment outcomes, involuntary hospital treatment takes longer, and IVT patients seem to struggle significantly more with weight restoration. We argue that more patient- and family-oriented treatment options, such as home treatment, might offer a promising approach to shorten or even avoid involuntary hospital admissions and further IVT. Different home treatment approaches, either aiming at preventing hospitalization or at shortening hospital stays, and the results of pilot studies are summarized in this article.

## 1. Introduction

Involuntary hospitalization is still a common practice, affecting almost one-quarter of children and adolescents admitted to a psychiatric hospital in high-income Western countries [1]. It is primarily applied to protect individuals unable to consent to treatment, usually for acute, life-threatening conditions or urgent treatment needs due to mental disorders [2]. However, the criteria for applying involuntary treatment (IVT) vary between countries and even within regions of the same country [1]. Although intended to protect patients or sometimes others, mostly carers, IVT raises numerous controversies. First, IVT is associated with ethical issues, as these health- and potentially life-saving interventions have repeatedly been reported to be potentially traumatic and harmful, inducing feelings of shame and guilt in patients [3]. Second, IVT is usually evaluated negatively, not only by patients themselves but also by professionals [4], relatives [5], and the general population. Furthermore, questions have been raised about the long-term effects of IVT, as a negative impact on trust, future help-seeking behavior, and engagement with medical staff has been reported [6]. Published interviews of adolescents and young adults who experienced IVT demonstrated that up to three-quarters of the interviewed patients reported distrust in medical staff as a direct consequence of IVT, including an unwillingness to seek help in the future even if feeling suicidal [6]. However, this was specifically the case for patients who retrospectively did not experience the hospital treatment as beneficial.

In child and adolescent psychiatry, IVT is regularly applied when treating severe cases of anorexia nervosa (AN) [7,8], a disorder marked by intense fear of gaining weight, a distorted body image, and behaviors leading to significant weight loss and severe health impairments without any insight into the seriousness of the illness [9]. The prevalence of AN among young individuals has increased with exacerbating numbers since the COVID-19 pandemic [10,11,12,13], especially in children below the age of 14 years [14,15]. AN is known to have the highest mortality among mental disorders, with rates 5 to 10 times higher among individuals with AN than among those of the age- and sex-matched healthy population [15], explained in part by suicide but mostly by medical complications due to malnutrition [16,17,18,19].

The earlier an adequate and evidence-based treatment is initiated for child and adolescent AN, the better the outcome will be [20]. Children and adolescents with AN are at an increased risk for devastating long-term consequences due to malnutrition, such as stunted growth, interrupted sexual development, abnormal cardiac function, and even dysfunctional brain maturation [21,22,23]. However, despite the profound impact of AN, children and adolescents often delay or even refuse intensive treatments such as (re)hospitalization [24]. Furthermore, treatment dropout rates are much higher compared to other mental disorders, with 20–53% of all inpatients with AN terminating treatment before completion [25]. This is especially daunting, as patients discharged prematurely against medical advice are at an increased risk of chronicity and adverse outcomes [26]. Factors such as the ego-syntonic nature of this disorder [27], impaired cognitive and decision-making abilities due to malnutrition [28], and the impact of separation from family and peers during hospital stays [24] contribute to these decisions.

The application of IVT for AN, especially within the child and adolescent population, has sparked intense debate for many years [29,30,31,32], rooted in ethical and legal principles, as empirical data were lacking. Ethical debates were centered on the principles of the patients’ “autonomy” versus “paternalism”, as well as the principles of “non-maleficence” (do not harm) versus “beneficence”. Concerning the debate about autonomy versus paternalism, it has, on the one hand, long been known that managing patients’ fluctuating compliance to treatment is central to AN therapy [33], and some degree of “force” even seems inevitable when treating patients with AN [5,34]. On the other hand, it has clearly been stated that even if patients are not fully capable of making reasonable decisions, IVT should only be applied in exceptional cases, and only if it is in the best interest of the patient. The second debate about “non-maleficence” versus “beneficence” focusses on the live-saving aspect of IVT in AN, as well as the fact that almost all the complications of AN are reversible with the early commencement of treatment, i.e., weight restoration, favoring IVT. While emphasizing patient autonomy and offering choices within treatment settings have shown potential to increase compliance and promote feelings of control and self-efficacy among patients with AN [32,35], the need for empirical data to understand the consequences and the potential harm of IVT in AN is essential. Suggestions for guidelines based on such empirical evidence are warranted, as in addition to varying legal preconditions, the attitudes and choices of physicians vary widely based on individual factors and knowledge of the disorder [36].

## 2. Methods

In this article, we aim to provide a narrative review of IVT within the field of AN. We opted for a narrative review to provide an update on the literature on this topic, specifically focusing on and summarizing the results regarding child and adolescent AN while emphasizing the clinical implementations. Clinicians are frequently directly involved in and confronted with decisions concerning IVT, especially within the field of child and adolescent AN. As all the authors of the current review are child and adolescent psychiatrists or psychotherapists working on a ward specialized in eating disorders, we aimed to use this empirical overview to focus on clinical implementations related to IVT in child and adolescent AN.

For this purpose, the PubMed, Web of Science, and Google Scholar databases were searched using the following terms: “treatment refusal”, “forced feeding”, OR “compulsory/coercive/involuntary/forced treatment/admission” AND “eating disorders” OR “anorexia nervosa”. Studies published from 2021 up to September 2023 were included in the search, as a systematic review and meta-analysis on compulsory treatments in eating disorders using the same search terms (also including the term “bulimia nervosa”) for earlier studies was published in 2021. The literature reported in this meta-analysis was also included [37]. Furthermore, additional articles were also found via hand searches of the reference lists of all the retrieved articles. For an overview of the included literature published from 2021 up to September 2023, see Table 1.

Subsequent to this update on empirical findings, this narrative review aimed to provide an overview of possible treatment advances as alternatives to hospital treatment to broaden the perspective and to subsequently move away from involuntary hospitalization within the treatment of child and adolescent AN.

## 3. Results

### 3.1. IVT in Child and Adolescent AN

When children and adolescents with AN do not agree to treatment despite threatening medical conditions, IVT frequently stands as a life-saving strategy [43]. Investigations about the prevalence rate of IVT for AN are scarce, and this is also because IVT might range from informal coercion in which patients comply with being treated because of persuasion by clinicians, family, and friends [24] to formal coercion determined by the court [44]. In a review from 2014 including data on adolescent and adult patients with AN, IVT was reported for 13 to 44% of all patients admitted to the hospital [8]. Even though, to our knowledge, there are no up-to-date reports on the rate of IVT specifically for child and adolescent AN, the literature generally demonstrates that IVT is more willingly applied to children and adolescents compared to adults [7]. Furthermore, within the population of patients with AN, young adolescents, in particular, perceive more coercion and less agreement than adults [24,45]. Unfortunately, patients are frequently in a much worse condition when IVT is initiated compared to the first referral for hospital treatment [8]. This might be the case because parents frequently underestimate or even deny the severity of the illness and thus do not grasp the necessity of intensive inpatient treatment (IP) [34]. Due to the patients’ serious medical states, when they are finally admitted involuntarily, hospital staff and parents might, in turn, have to consider the application of further coercive interventions such as the restriction of physical activity, forced tube feeding, forced medication, or even physical restraint [7,42]. Even though there are some studies focusing on different kinds of IVT, unfortunately, to our knowledge, none of these studies has differentiated between the effects of different measures of IVT on children or adolescents with AN. Thus, no conclusion about the exact effects of specific interventions can be drawn.

An explorative study based on qualitative interviews reported that even coercion within legally voluntary treatment can cause patients to experience excruciating feelings of pressure and guilt [46]. The same investigation highlighted the potential traumatic risks of coercion, including physical restraint [46]. A recent investigation on the impact of forced tube feeding on patients, their carers, and medical staff also highlights forced tube feeding as traumatizing for everyone involved [41]. However, it was acknowledged by all involved that this intervention was potentially lifesaving, with the short-term benefit of medical stabilization and the long-term reflection that this intervention may have been a turning point in the patient’s treatment journey, which allowed for full recovery [41].

#### 3.1.1. Comparison of Patient Characteristics at Admission—Voluntary vs. Involuntary Admissions

Considering possible differences in patient characteristics at admission when comparing voluntary and involuntary admissions, findings across the whole age range of patients with AN are mixed. A systematic review from 2014 [8], including studies on adolescents and adult patients, reported a higher number of psychiatric comorbidities, higher incidences of self-harming behavior and a longer duration of AN in patients admitted involuntarily, but no differences were reported regarding the somatic condition between those who were admitted voluntarily and those who were admitted involuntarily. It was therefore suggested that IVT is not only initiated due to the severity of the somatic symptoms alone but also presumably as a response to the complexity of a patient’s situation. A more recent systematic review and meta-analysis [37] did, however, report a significantly lower BMI in patients admitted involuntarily. However, even though three rather old studies included in this meta-analysis reported a longer duration of AN [47,48,49], due to opposing results in other more recent studies [50,51], the result of this meta-analysis [37] showed no statistical significance regarding illness duration in the patients admitted involuntarily. In addition to these findings, a recent retrospective study found that the binge eating/purging subtype of AN and a higher number of previous admissions for AN were the strongest predictors of IVT in patients with AN [38].

For adolescents, the literature is especially scarce. The results of the only study of IVT solely including adolescent patients suggested that the duration of AN and the early onset of the disorder are significant predictors for IVT [26]. Furthermore, adolescent patients admitted involuntarily showed reduced psychosocial functioning, a higher level of comorbid depression, and a higher rate of suicidal behavior [26], complying with the hypothesis that IVT occurs as a response to the complexity of multiple symptoms rather than due to the severity of the symptoms alone [52]. Another study on perceived coercion in adolescent patients with AN reported more perceived coercion in patients with a higher weight and less severe eating disorder psychopathology, reflecting a subjectively perceived lower need for hospitalization [45]. These results also contradict the hypothesis that the severity of the symptoms alone predicts IVT or perceived coercion.

#### 3.1.2. Comparison of Treatment Outcomes—Voluntary vs. Involuntary Admissions

In addition to the question of whether patient characteristics differ between involuntarily and voluntarily admitted patients, questions have been raised about the treatment consequences and effects of IVT. This is especially interesting, as motivation to change is known to be an important predictor for successful treatment in AN patients [45,53] and is expected to be lower if treatment is involuntary. However, study results for adult as well as adolescent patients with AN all suggested that the overall effect of IVT is comparable to that of voluntary treatment regarding treatment outcomes [8,26,37]. First, similar results on weight gain and BMI at discharge were reported across all studies [8,37,39], with one earlier study from 2000 even reporting greater weight gain for involuntarily admitted patients [50]. These results demonstrate that IVT does not seem to be inferior to voluntary treatment regarding weight gain and that the potentially lower weight at admission is successfully increased. Moreover, in the one study examining only adolescent patients, it was even shown that patients who were admitted involuntarily showed a greater improvement in their psychopathology, and regular menstruation was resumed in significantly more patients in the detained group at discharge [26], an important measure of recovery in AN patients. Only a higher number of psychiatric comorbidities was associated with a lower success rate of IVT [42]. This last finding is not surprising, as psychiatric comorbidities in patients with AN are known to potentially delay or make the therapeutic process impossible, to burden the patient even more, and might destroy his/her compliance with treatment.

#### 3.1.3. Comparison of Long-Term Effects—Voluntary vs. Involuntary Admissions

Regarding the negative long-term effects as well as the high mortality rate in patients with AN, follow-up data comparing patients admitted to hospital involuntarily compared to those admitted voluntarily are of essential interest. The results of a meta-analysis [37], including three studies that assessed follow-up data [26,47,54], found no differences in mortality risk between voluntarily and involuntarily admitted patients [37]. A very recent study confirmed these results and further reported no differences between involuntarily and voluntarily admitted patients regarding BMI, readmissions, or quality of life on average 4 years after discharge [39]. In a previous study, Ayton et al. (2009) even showed significantly better long-term outcomes for adolescent patients treated involuntarily, including better general functioning and normalization of weight [26].

#### 3.1.4. Comparison of the Treatment Process—Voluntary vs. Involuntary Admissions

Although the reported results above suggest a lack of worse outcomes of IVT in patients with AN, especially in adolescents, it was constantly reported that treatment under involuntary conditions took longer, depriving patients of their natural social environment for longer time periods [8,37]. Furthermore, the treatment of patients admitted involuntarily involved higher rates of nasogastric feeding [8,26], refeeding syndrome [55], and psychotropic medication use to alleviate patients’ distress as well as comorbid symptoms [26]. Thus, patients who were admitted involuntarily not only seemed to be in a slightly worse medical condition at the time of admission but also seemed to struggle significantly more during weight restoration. This might, in part, be explained by the suggested postponement of hospital admission and the related worsening and chronification of the symptoms [8], as admission to hospital alone is experienced as drastic and coercive [24]. Furthermore, it should be noted that none of these studies differentiated between different “severity levels” of coercive interventions. It can be hypothesized that physical restraint might be experienced as especially traumatic, potentially leading to different treatment outcomes. However, research concerning the type, frequency, and effects of the use of physical restraint for adolescent AN is especially scarce. According to our search, only one study reported on physical restraint during the IP of adolescents with AN [56]. In accordance with the reported results of the current review, at the 5-year follow-up, the authors found no significant differences in readmission rates, BMI, or ED psychopathology between patients with and without physical restraint episodes. However, they reported that within their study population, only a very small number of patients accounted for most of the reported restraint episodes [56], which was also reported in another cohort study on IVT [57]. Additional investigations of patients with frequent physical restraint episodes seem to be necessary to draw any conclusions on the effects of this type of coercive intervention.

In addition to the known general negative experience of coercive treatment procedures [3,6], as well as the related ethical and legal issues [29,31,32,42], even though treatment outcomes might be favorable, long-lasting hospitalizations contribute to delayed development during adolescence and severe social impairments in patients with AN [58,59]. Therefore, alternative treatments to hospitalization are still necessary, not only to improve the prognosis for patients with AN but also to offer alternatives to hospitalization that might be less drastic and coercive, preferably also resulting in less postponement of necessary intense treatment and preventing the related worsening of patients’ conditions. This might further help to prevent the necessity of IVT.

### 3.2. Alternatives to Hospitalization in Child and Adolescent AN

In several European countries, IP is considered the gold standard when treating children and adolescents with severe AN [60]. IP is often perceived as coercive because hospital stays are particularly long for patients with AN, at least in several European countries [61]. An analysis of AN patients’ attitudes toward compulsory treatment emphasized that the anticipated long duration of IP is a significant factor in experienced compulsion at admission [46]. Furthermore, it has repeatedly been found that within 1 year after discharge for a prior hospital intervention, up to one-third of adolescents with AN are readmitted for IP [62,63], questioning the effectiveness of long IP stays. One explanation might be that while often effective for weight stabilization, IP frequently fails to successfully modify specific eating-disordered (ED) behavior (e.g., the “drive for thinness”, [64]). In addition, long-lasting IP separates adolescents from their peers and family, contributing to impairments in the social development of young patients with AN [58]. Thus, to avoid coercion as well as experienced pressure and to establish more or at least similarly effective treatment strategies, alternatives to long IP are urgently needed.

Within recent years, there have been several attempts to prevent or at least shorten inpatient admissions for children and adolescents suffering from AN [65]. One attempt is to focus on improving outpatient treatment in order to prevent inpatient admissions, aimed at directly involving the patients’ caregivers into the treatment process [66]. Currently, a large body of evidence supports family-based treatment (FBT), which gives patient caregivers the lead in refeeding their child, which is one of the most successful outpatient treatment approaches applied in children and adolescents with AN [67,68,69,70]. However, evidence is still limited regarding the outcomes of patients following FBT; depending on the readmission criteria, fewer than 40% of children and adolescents are remitted at the end of a standard FBT and eventually require a more intensive treatment, such as IP settings [66].

Another attempt aimed at shortening hospital stays in child and youth with AN and facilitating earlier and more intensive family involvement is the so-called “stepped-care” approach, which involves a gradual transfer from IP settings of several weeks to less intensive treatment settings such as day patient treatment [71,72]. The more positive outcome of “stepped-care” treatment compared to IP alone was ascribed to earlier and more intensive family involvement, as well as earlier active practice within environments familiar to the patients [73]. Furthermore, by shortening inpatient stays, an initial short IP might appear less coercive, potentially preventing patients from postponing or even rejecting inpatient admission. However, even though study results present a slight superiority to IP regarding specific treatment outcomes [72,74], 30% of the day treatment patients included in the largest RCT with adolescent AN patients had to be re-hospitalized afterwards [72]. Patients and their parents did not feel prepared to handle the ED in their home environment, even though early family involvement was suggested to be a successful therapy component.

Considering the promising effects of early and more intense family involvement and the practice of healthy habits within the direct patient environment under the supervision of the parents of adolescent patients, the need to establish specific home treatment approaches for treating child and youth with AN has been repeatedly noted [72,75]. Efforts to develop and implement specific home treatment programs for treating mental disorders go back into the 1980s (e.g., [76]). Alongside the aim to reduce the burden of long IP for individuals, families, and society, it was considered to be especially important to reduce the negative consequences of long-term hospital stays for young children and adolescent populations [59]. Home treatment is defined as an intensive treatment modality with the aim of providing direct therapeutic support within the homes of patients with a mental disorder [77]. While emphasizing the involvement of family members, frequent and regular visits to the patient’s home are conducted by a multidisciplinary treatment team [77]. These multidisciplinary teams generally involve psychiatrists and/or psychologists, occupational therapists, nursing staff, and social workers.

### 3.3. Home Treatment for Child and Adolescent AN

In recent years, specific home treatment programs have been developed in the field of child and adolescent AN [71,78,79,80,81]. These programs either aim at preventing hospitalization (augmented outpatient interventions) [78,79,80,81,82] or at shortening hospital stays (stepped-care interventions) [71]. Focusing on preventing (in)voluntary hospital admissions, different research groups around the world are aiming to improve the existing outpatient approach and develop home treatment programs as supportive add-ons to outpatient FBT [77,78,79,80,81,82]. The main aim of these “add-on home sessions” is to provide practical support for caregivers. This treatment strategy reinforces parental resources for weight restoration as well as for disrupting typical ED behavior. Furthermore, by directly supporting patients in their social environment, the home treatment sessions aim at promoting patient and family resilience factors and resources (e.g., re-engagement with hobbies, meeting and eating with friends). A recently published pilot study on this approach, using a waitlist control design, showed that home treatment as an add-on to FBT seemed to be a feasible and suitable method for treating AN in adolescents [79]. It was found that both groups (FBT-only vs. FBT + home treatment) showed significant weight gain as well as a reduction in the number of patients meeting the diagnostic criteria for AN at the end of treatment. Furthermore, the “FBT + home treatment” group showed a significantly greater increase in BMI. Menses resumed in these patients more frequently, and no patients had to be transferred to the hospital during the treatment period (compared to 13.6% in the FBT-only group).

The Department of Child and Adolescent Psychiatry at the University of Aachen, Germany, focused on shortening hospital stays and developed a “stepped-care” home treatment program for adolescent patients with severe AN [71,77]. Patients first undergo an inpatient stabilization period of 5–8 weeks followed by 12–16 weeks of a manualized home treatment program with a declining frequency of home visits, at least one family session per week, and an additional group therapy session. The main aim of this stepped-care treatment approach, including home-based therapy, is to offer more direct and especially individualized support within the home environments of patients with severe AN and their caregivers. It should thus interrupt the vicious cycle of ED behavior and weight loss and therefore facilitate reintegration into age-appropriate activities. A recently published pilot study on this approach including 22 patients with severe AN who were admitted for IP demonstrated that this stepped-care approach was feasible and safe for adolescent patients with AN and thus considered as a promising new approach to reducing the duration of hospitalization for IP and, furthermore, to potentially improving the longer-term outcomes for this patient population [71]. It was found that all patients completing the home treatment program not only showed a significant improvement in their ED behavior and general psychopathology but, moreover, achieved significant weight gain, which was maintained one year after hospital admission [71,83]. Furthermore, all patients reported high treatment satisfaction, and their motivation to change increased significantly during the treatment and remained stable at the 1-year follow-up [71,83]. Additionally, the ability of caregivers to handle their children’s ED significantly improved throughout treatment, while their perceived burden and depressive symptoms decreased [71,84]. Currently, a large multicenter randomized controlled trial is being conducted to verify these pilot results (funded by the Federal Joint Committee, Germany (G-BA innovation fond), grant number 01VSF20006).

## 4. Conclusions

This review article summarized the scarce empirical data comparing voluntary and involuntary treatment in patients with AN with a special focus on children and adolescents, suggesting no specific inferiority of IVT regarding the specific treatment outcomes at different follow-up points [8,26,37,39,50]. However, involuntary hospital treatments have been shown to take longer [8,37], and patients admitted involuntarily seem to struggle significantly more during weight restoration, with a higher rate of nasogastric feeding [8,26], a higher rate of refeeding syndrome [55], and increased use of psychotropic medications [26]. Additionally, the binge eating/purging subtype of AN and a higher number of previous admissions for AN were shown to predict IVT in patients with AN [38]. Furthermore, it has been shown that the presence of significant psychiatric comorbidities may modify outcomes and negatively influence the success of IVT [42,85].

One possible explanation for the lack of difference in treatment outcomes following involuntary hospitalization compared to voluntary hospitalization might be that due to the ego-syntonic nature of AN, voluntarily admitted patients also experience hospital admission or treatment as coercive [24,27] but still comply ostensibly. On the other hand, even though IVT is generally evaluated negatively, it has been shown that attitudes toward IVT often change over time during or even after treatment, with involuntarily admitted patients retrospectively recognizing the need for treatment [24,40,56]. This was also supported by a recently published case report highlighting the relief and reduced feelings of guilt and anxiety related to the choice to receive nutrition on the part of patients receiving involuntary tube feeding [86]. However, a recent study on the individual perspectives of patients highlighted that this was only the case for patients reporting favorable changes in their ED symptoms [87]. Patients whose perspectives about IVT remained negative showed no changes in their ED recovery post treatment [87].

There are several limitations to our review. First of all, we could not cite any current prevalence rates of IVT. Unfortunately, to our knowledge, there are no national reports on recent rates of coercive treatment for juvenile AN nor any reports across countries. Secondly, empirical data on IVT in children and adolescents are especially scarce, including retrospective analyses. Thirdly, the empirical data summarized within the current article did not specifically investigate the consequences of different elements of IVT, such as physical restraint, as this coercive treatment procedure might be experienced as especially traumatic, potentially leading to different treatment outcomes, as well as long-term effects. Thus, alongside the urgent need for recent prevalence rates of IVT, as well as recent empirical data on the consequences of IVT, especially within the field of child and adolescent psychiatry, more specific investigations of the different components of IVT, such as retrospective analyses, seem necessary to draw any conclusions on the effects of these coercive interventions and might reveal differences regarding treatment outcomes in the future.

Another very important aspect that should be empirically investigated in young patients with AN is the impact of coercive approaches on young patients’ trust and thus their future help-seeking behavior and engagement with medical staff later in life. This negative impact has already been reported in a general psychiatric population [6] and might be expected to also apply to patients with AN, even though comprehensive studies are lacking and should thus be performed in the future. This is of particular interest, as it is known that a large portion of patients suffering from child or adolescent AN might develop a chronic disease and are at a high risk of receiving a diagnosis of another psychiatric disorder in subsequent years, demanding further professional help seeking [88]. Early treatment with good compliance and cooperation might therefore pave the way for the adequate use of health care in the future.

To summarize, in many cases of severe AN, involuntary treatment seems inevitable and does not result in inferior treatment or long-term effects when compared to voluntary treatment; however, it still has a variety of adverse effects, especially during the treatment process itself. On the one hand, these findings should encourage clinicians to step away from the idea that IVT always causes harm and is generally inferior to voluntary treatment. However, on the other hand, the potentially harmful experiences of IVT, combined with the knowledge that the “gold standard” of IP for children and adolescents with severe AN in most European countries [60] seems questionable regarding its general effectiveness [62,63,89], demonstrate the need to identify alternative treatment approaches for child and adolescent AN. Thus, possible treatment advances in alternatives to hospital treatment, in order to move away from coercive treatment for child and adolescent AN, should be investigated in the future. Among other treatment options, such as day patient treatment, home treatment might offer a promising new and alternative treatment approach for children and adolescents with severe AN to shorten [71] or even avoid (involuntary) hospitalization stays [78,79,80,81,82]. Pilot studies of both approaches thus far suggest that these approaches provide patients and their families with the right to make choices, thus strengthening their autonomy in the treatment process. However, the data are preliminary and scarce, and there is an urgent need to confirm the presented pilot data using larger samples.

## Figures and Tables

**Table 1 healthcare-11-03149-t001:** Characteristics and main findings of included studies on involuntary treatment in anorexia nervosa.

Study (Ref.)	Study Design	N	Diagnosis	Age Group	Method(s) of IVT	*Outcome Measures* & Main Findings
Atti, A.R., et al., 2021 [37]	systematic review & meta-analysis	N = 242 IVT patients N = 738 VT patients	AN, BN, EDNOS	adolescents (1 study) & adults (8 studies)	involuntary hospital treatment	- *BMI at admission*: lower BMI in IVT patients - *length of hospitalisation*: longer in IVT patients - *BMI at discharge, illness duration, mortality:* no differences between groups
Di Lodovico, L., et al., 2021 [38]	retrospective study	- N = 36 IVT patients - N = 72 VT patients	AN	adults	involuntary hospital treatment	- *BMI:* history of lower weight for IVT patients - *previous admissions for AN*: more for IVT patients - *psychotropic medication*: more frequently prescribed for IVT patients - *socio-economic status:* lower for IVT patients - *length of hospitalisation:* longer in IVT patients - *type of AN*: more binge eating/purging subtype
Ramasamy, R.S., 2021 [7]	case study	N = 1 IVT patient	AN	adolescent	involuntary hospital treatment	By removing the responsibility of decision-making from both the patient & the family by IVT, recovery can be initiated even in severe and enduring AN.
Abry, F.P., et al., 2023 [39]	retrospective longitudinal study	- N = 23 IVT patients (M_follow-up_: 2.7 years) - N = 25 VT patients (M_follow-up_: 5.6 years)	AN	adults	involuntary hospital treatment	- *BMI at follow-up:* lower for IVT patients - *readmissions, quality of life, mortality*: no differences at follow-up - *perceived need for hospitalisation:* no differences at follow-up (improvement over time in IVT patients)
Mac Donald, B., et al., 2023 [40]	Qualitative interview study	N = 7 IVT patients	AN	adults	involuntary hospital treatment, mechanical restraint, physical restraint, involuntary NGT feeding, constant observation	- IVT can help an internal battle against AN - perspectives of patients about IVT can change over time - IVT can have a negative impact on patients, such as feelings of being hunted or assaulted
Fuller, S.J., Tan, J., & Nicholls, 2023 [41]	Qualitative interview study	N = 7 IVT patients N = 13 parents/carers of IVT patients N = 16 clinical staff	AN	patients = adults parents/carers of adolescent & adult patients clinical staff of adolescent & adult patients	involuntary NGT feeding	- acknowledgement that involuntary NGT feeding can be lifesaving (short term benefit: medical stabilisation & long-term benefit: turning point to fully recover) - involuntary NGT feeding is traumatising to all involved
Tumba, J., Smith, M., & Rodenbach, K.E., 2023 [42]	perspective on clinical and ethical dilemmas	/	AN	/	involuntary hospital treatment	/
Clausen, L & Jones, A., 2014 [8]	systematic review	- N = 231 IVT patients - N = 642 VT patients	AN	adolescents & adults	involuntary hospital treatment, involuntary NGT feeding	- IVT patients were characterised by a more severe “psychiatric load” at admission (higher co-morbidity, more preadmissions, longer duration of illness and more incidences of self-harm) - no differences between groups regarding the levels of eating disorder pathology & the outcome of treatment in terms of symptom reduction
Ayton, A.C., Keen, C., & Lask, B., 2009 [26]	naturalistic study	- N = 16 IVT patients - N = 34 VT patients	AN, EDNOS	adolescents	involuntary hospital treatment	- *age of onset*: earlier in IVT patients - *previous admissions for AN:* more for IVT patients - *psychosocial functioning at admission*: worse in IVT patients - *comorbid depression & suicidal behaviour at admission*: more in IVT patients - *length of hospitalisation*: longer in IVT patients - *NGT feeding:* more frequently in IVT patients - *psychotropic medication*: more frequently prescribed for IVT patients *- BMI at discharge & all psychological measures*: no differences between groups

IVT = involuntary treatment, VT = voluntary treatment, AN = anorexia nervosa, BN = bulimia nervosa, EDNOS = eating disorder not otherwise specified, NGT = nasogastric tube.

## Data Availability

Not applicable.

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
