# Peer review of "Involuntary Treatment for Child and Adolescent Anorexia Nervosa—A Narrative Review and Possible Advances to Move Away from Coercion"

_healthcare, 2023, doi:10.3390/healthcare11243149_

Round 1

Reviewer 1 Report

Comments and Suggestions for Authors

Please consider explaining why a narrative review approach was an appropriate choice for your method. Also consider adding a little about the background of the researchers as I think knowing the professional contexts of the researchers in relationship to the study gives us readers some important insights.

Author Response

Reviewer 1:

  1. Please consider explaining why a narrative review approach was an appropriate choice for our method. 
    # 1: The reviewer is right that we did not explain well enough why a narrative review on this subject was chosen. We added more elaborate information to the Method section (p. 3, ll. 132-139) and hope that Reviewer 1 agrees with our explanation. Additionally, we added some specific clinical advice to the conclusion section (p. 9, ll. 458-460).

  1. Also consider adding a little about the background of the researchers as I think knowing the professional contexts of the researchers in relationship to the study gives us readers some important insights.

# 2. We also agree with the reviewer that this information gives some important insights, also regarding the choice of providing a narrative review on this topic. We have thus added the information that all authors are child and adolescent psychiatrists or psychotherapists working at a specialized eating disorder department to the method section (p. 3, ll. 136-139).

Reviewer 2 Report

Comments and Suggestions for Authors

This narrative review on Involuntary Treatment for Child and Adolescent Anorexia 2 Nervosa addresses an important and interesting topic. However, few concerns should be addressed prior to publication. Here are my comments:

1. The introduction is informative, but it could be more concise. It should be condensed to clearly convey the essential background information without overloading the reader with details.

2. When citing statistics or previous research, ensure that the sources are up-to-date. Make sure to use the most recent statistics and studies available to enhance the relevance of the review.

3. Discuss ethical considerations regarding involuntary hospitalization in more detail. What ethical principles are at stake, and how do they relate to patient autonomy? Expanding on this point will help readers better understand the ethical context.

4. The section on treatment outcomes is somewhat convoluted. Consider restructuring it to provide a clearer delineation of the findings and their implications. Separate findings on involuntary vs. voluntary hospitalization and clarify the differences.

5. Elaborate on the impact of psychiatric comorbidities in AN patients. How do these comorbidities affect treatment outcomes, especially in the context of involuntary hospitalization?

6. Address the potential long-term effects of involuntary hospitalization more comprehensively. Discuss how this might influence a patient's future help-seeking behavior and the overall engagement with medical staff.

7. While discussing the rates of involuntary treatment for child and adolescent AN, try to provide recent and specific data if available. This will help readers understand the current landscape.

8. To provide a balanced perspective, include a section that directly compares the effectiveness of involuntary treatment with voluntary treatment. Provide data that allow readers to draw a clear conclusion about the relative benefits and drawbacks.

9. The manuscript mentions the use of coercion, including physical restraint, but lacks in-depth analysis. Further investigate the type and frequency of coercive interventions to evaluate their specific effects on treatment outcomes.

10. In the conclusion, suggest areas for future research in more detail. What specific questions remain unanswered, and what new research avenues should be explored in the context of AN treatment?

11. Consider adding tables to the manuscript to summarize results and/or to add more comprehensive results as necessary. A narrative review cannot be published with comprehensive tables on the results.

Comments on the Quality of English Language

Minor editing is required.

Author Response

Reviewer 2:

Quality of English Language

(x) Minor editing of English language required

The paper was evaluated and corrected by a native speaker prior to submission using the editing assistance by AJE (American Journal Experts). In addition, we thoroughly checked spelling, grammar, and punctuation again and identified a few mistakes which were corrected throughout the manuscript. However, due to the suggestions of the reviewer, multiple adaptations have been made. If the reviewer and/or editor suggest a further correction, we are willing to re-submit it to AJE.

  1. The introduction is informative, but it could be more concise. It should be condensed to clearly convey the essential background information without overloading the reader with details.

# 1. According to the reviewer’s suggestion we shortened our introduction, and, to our mind, it is more concise now. However, according to the reviewers’ third comment, we have also added some details regarding the ethical considerations. We hope that by having shortened parts of the introduction and added details concerning the ethical debate, we were able to comply with both valuable suggestions of the reviewer.

  1. When citing statistics or previous research, ensure that the sources are up-to-date. Make sure to use the most recent statistics and studies available to enhance the relevance of the review.

# 2. We agree with the reviewer that up-to-date literature is essential to enhance the relevance of an article. However, we tried to find more current literature and have used most recent articles on the current topic. Unfortunately, some of the literature on coercive treatment, especially within the field of child and adolescent AN is rather ancient and no recent alternatives are available. As we agree that this is a limitation of the current review, we have added this limitation to the conclusion section more specifically (p. 9, ll. 426-430).

  1. Discuss ethical considerations regarding involuntary hospitalization in more detail. What ethical principles are at stake, and how do they relate to patient autonomy? Expanding on this point will help readers better understand the ethical context.

# 3. The reviewer is right that adding further information to the ethical considerations will help the reader to better understand the ethical context. We thus have done accordingly (p. 2, ll. 96-98 & p. 3, ll. 105-108).

  1. The section on treatment outcomes is somewhat convoluted. Consider restructuring it to provide a clearer delineation of the findings and their implications. Separate findings on involuntary vs. voluntary hospitalization and clarify the differences.

# 4. We thank the reviewer for this important point. We considered reporting the findings on voluntary and involuntary treatment separately. As we already contrast the findings regarding different phases of treatment and do think that this choice of contrasting does have its benefits, we have decided to include sub-headings to improve readability and to provide additional structure to this section. We do hope that the reader is guided through the findings more clearly now and hope that the reviewer is satisfied with our solution.

  1. Elaborate on the impact of psychiatric comorbidities in AN patients. How do these comorbidities affect treatment outcomes, especially in the context of involuntary hospitalization?

# 5. The reviewer is right, that the reported finding on psychiatric comorbidities was not clear, as we were only using the word “comorbidities”. The reported finding, that a high number of comorbidities in AN patients was associated with a lower success rate of IVT, was however specifically found for psychiatric comorbidities. We have therefore first of all simply added the word “psychiatric” (p. 5, l. 233). Secondly, we have added a potential explanation for this reported finding (p. 5, ll. 234-237). We hope that the reviewer is satisfied with this solution.

  1. Address the potential long-term effects of involuntary hospitalization more comprehensively. Discuss how this might influence a patient's future help-seeking behavior and the overall engagement with medical staff.

# 6. According to the reviewer’s suggestion we have added some further information on the patients´ interviews of their future help-seeking behavior (p. 2, ll. 47-52).

  1. While discussing the rates of involuntary treatment for child and adolescent AN, try to provide recent and specific data if available. This will help readers understand the current landscape.

# 7. We agree with the reviewer that our review would benefit strongly from recent and specific data of involuntary treatment for child and adolescent AN. However, we have really tried to get some newer data and even referred to the German justice and youth welfare system. To the best of our knowledge, current data are not available. Also, regarding comment “10” we have added this information and the need for specific numbers on this matter to the conclusion (p. 9, ll. 426-428).

  1. To provide a balanced perspective, include a section that directly compares the effectiveness of involuntary treatment with voluntary treatment. Provide data that allow readers to draw a clear conclusion about the relative benefits and drawbacks.

# 8. To the best of our knowledge, section 3.1.2 does provide the studies which directly compare the effectiveness of IVT with VT. Due to the narrative character of the review, no meta-analysis is provided. Please note however that it is extremely difficult to directly compare IVT with VT patients, as their severity of illness often differs.

  1. The manuscript mentions the use of coercion, including physical restraint, but lacks in-depth analysis. Further investigate the type and frequency of coercive interventions to evaluate their specific effects on treatment outcomes.

# 9. We do agree with the reviewer that a further differentiation of the different types of IVT would be desired, also for individual therapeutic planning. Unfortunately, and we point this out already (p. 9, ll. 430-436), a clear differentiation is often not possible in practice, also due to the complex treatment approaches which comprise physical activity, re-nutrition, psychotherapy, and weight restoration. In any case, even though there are some studies focussing on different types of IVT (e.g. involuntary nasogastric tube feeding, involuntary hospital admission), to our knowledge, none of the studies has differentiated between the effects or even the frequency of different measures of IVT, especially not in children and adolescents with AN. We state this now more elaborately (p. 4, ll. 174-178) and ask for these types of studies in the conclusion section in the future (p. 9, ll. 434-438).

  1. In the conclusion, suggest areas for future research in more detail. What specific questions remain unanswered, and what new research avenues should be explored in the context of AN treatment?

# 10. According to the reviewers’ suggestion, we have elaborated on the different suggestions for future research in more detail (ll. 434-438, ll. 440-441 & ll. 444-445).

  1. Consider adding tables to the manuscript to summarize results and/or to add more comprehensive results as necessary. A narrative review cannot be published with comprehensive tables on the results.

# 11. We agree with the Reviewer, that the review benefits from an overview of the included literature and the reported results. Also, in line with one of the editors’ comments, we have therefore included a respective table (Table 1).

Round 2

Reviewer 2 Report

Comments and Suggestions for Authors

Thank you for revising the manuscript and providing response to my comments.